# LLGformer: Learnable Long-range Graph Transformer for Traffic Flow Prediction

## Abstract

Traffic prediction plays a pivotal role in intelligent transportation systems. Most existing studies only predict traffic flow for a specific time period based on traffic data from a short period, such as an hour, overlooking the influence of periodicity present in traffic data. Moreover, most of the existing advanced methods rely on manually constructed spatio-temporal graphs for joint modeling, or use pure spatial and pure temporal modules to separately model spatial and temporal features, which limits the learning of complex spatio-temporal patterns in traffic data due to structural inadequacies in the model. To address these issues, we propose a novel approach by constructing a learnable long-range spatio-temporal graph, which can better capture complex patterns in traffic data. We introduce a new model, LLGformer, which improves upon traditional Transformer-style models, facilitating more efficient learning of traffic flow data by integrating long-range historical information. Leveraging attention mechanisms on a spatiotemporal graph enables direct interaction of information across different time slices and locations. Additionally, we propose two optimization strategies to further boost the speed of training and inference. Extensive experiments on four real-world datasets show that the new model significantly outperforms state-of-the-art methods.

## CCS Concepts

• **Information systems → Spatial-temporal systems**; • **Computing methodologies → Neural networks**.

## Keywords

Traffic flow prediction, Transformer, Predictive model, Spatio-temporal graph

**ACM Reference Format:**
Anonymous submission. 2018. LLGformer: Learnable Long-range Graph Transformer for Traffic Flow Prediction. In *Proceedings of Make sure to enter the correct conference title from your rights confirmation emai (WWW '25)*. ACM, New York, NY, USA, 11 pages. https://doi.org/XXXXXXX.XXXXXXX

## 1 Introduction

In recent years, urban traffic management has faced significant challenges due to population growth and the continuous rise in transportation demands. Intelligent Transportation Systems (ITS) [47], as a crucial component of modern smart cities, have been widely developed and applied to address external challenges and provide energy-efficient transportation solutions. Traffic flow prediction, one of the core tasks of ITS, aims to accurately forecast future traffic conditions through the collection and learning of historical traffic data [18]. Accurate traffic flow prediction can contribute solutions to various traffic management challenges [38], including traffic flow control [17, 46], route planning [13], and congestion alleviation [53].

Traffic prediction aims to forecast future traffic conditions based on the historical traffic conditions on urban roads. The most fundamental challenge in traffic flow prediction is how to effectively capture and model the spatio-temporal dependencies of the complex dynamics in traffic data [48]. A classic approach treats the traffic prediction problem as a time series forecasting problem [9, 35, 42]. Integrating deep learning methods with time series models such as Recurrent Neural Networks (RNNs) [11, 36] and Convolutional Neural Networks (CNNs) [20] enables the modeling of temporal dependencies. For instance, DeepAR [30] conducts probabilistic predictions in the form of Monte Carlo samples and optimizes model parameters, incorporating RNN to simulate the probability distribution of future sequences. The recurrent structure of RNNs significantly increases computational costs, which can be effectively addressed by using Temporal Convolutional Networks (TCNs). The application of attention mechanisms enhances the model's efficiency in learning time series, allowing it to capture long-range temporal dependencies between different timestamps [23]. Transformer [37, 40] serves as a prominent example of this approach. TPA-LSTM [33] combines attention mechanisms to selectively weight relevant variables, enabling the modeling of multivariate time series prediction problems. Pyraformer [27] introduces a Pyramid Attention Module (PAM) capable of capturing time dependencies at different ranges and resolutions. These classical time series prediction methods predominantly focus on temporal correlations and may struggle to handle complex and diverse spatial correlations that involve various long-range dependencies.

Currently, a more prevalent and effective approach is to consider the traffic prediction as a spatio-temporal graph modeling problem, where all sensors in the traffic network constitute a spatio-temporal graph to represent the continuously changing traffic conditions on city roads [18, 40]. In these methods, spatio-temporal graphs are used to represent the spatio-temporal correlations of time series data with a non-Euclidean spatial structure for predicting future traffic conditions. The emergence of Graph Neural Networks (GNNs) has led to the application of Spatio-temporal Graph Neural Networks (STGNNs) in traffic prediction tasks [12, 15, 39]. For example,STSGCN [34] connects individual spatial graphs of adjacent timestamps into one graph, effectively capturing continuous local spatio-temporal correlations. STFGNN [24] considers fusion operations of both time and space when constructing the spatio-temporal graph, utilizing stacked spatio-temporal fusion graph neural layers to learn spatio-temporal dependencies. TraverseNet

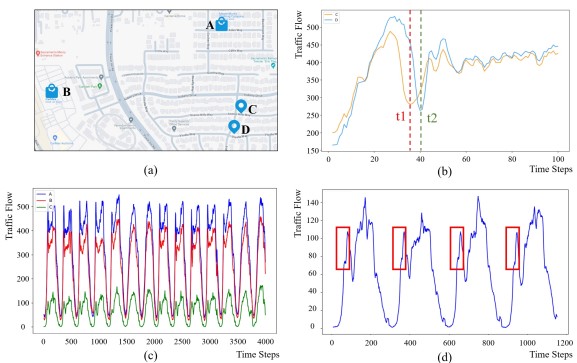

(a)

(b)

(c)

(d)

**Figure 1: (a) shows the road map of a certain area. (b), (c), and (d) display the traffic data variations recorded by four sensors, A, B, C, and D, during different time periods.**

[45] employs an attention mechanism on the spatio-temporal graph to selectively choose neighborhoods that influence nodes, utilizing a message traversal mechanism to learn past information points for each node and current neighborhood information. DSTAGNN [22] uses different self-attention modules to capture temporal and spatial information respectively. While existing methods effectively model spatio-temporal correlations, they still face many limitations.

First, existing models lack effective methods for constructing spatio-temporal graphs. In a traffic network, a node's current state can be influenced by its own state and those of neighboring nodes in the recent past. As shown in Figure 1(a), the road map highlights the geographical locations of four areas: $A, B, C$, and $D$. For example, Figure 1(b) illustrates how congestion at position $C$ at time $t_1$ may lead to congestion at nearby position $D$ at a later time $t_2$. Moreover, geographically distant regions with similar urban functions can exhibit analogous traffic patterns. Figure 1(c) demonstrates that despite the distance between $A$ and $B$, their traffic conditions show remarkably similar trends. This suggests that dependencies in location and time are often long-range rather than local.Models like STSGCN and STFGNN typically construct graphs with only three time steps, necessitating multiple hops or iterations to capture spatio-temporal dependencies, which can hinder performance. Therefore, it is essential to build a global spatio-temporal graph that facilitates direct interactions between data at any timestamp and any node. Trafformer [19] adopts a global perspective by utilizing a pre-built adjacency matrix for the spatio-temporal graph. However, this pre-built approach may not be efficient for downstream tasks.

Second, existing models underutilize historical information. Figure 1(c) illustrates traffic flow variations in regions $A$, $B$, and $C$ over two weeks, revealing clear periodic patterns, with similar traffic trends occurring at the same time of day on different dates. Most current methods predict traffic based solely on data from the previous hour, neglecting these periodic patterns. For example, when forecasting traffic from 10:00 AM to 11:00 AM, existing approaches often rely on data from 9:00 AM to 10:00 AM of the same day. However, as shown in Fig. 1(d), the data for this hour is highly correlated with data from 10:00 AM to 11:00 AM on previous days.

Finally, limitations exist in the model architecture design. Transformer-style encoder-decoder models are widely used in spatio-temporal

traffic prediction and have demonstrated strong experimental results. However, the autoregressive nature of the Transformer decoder means that errors at each timestamp can propagate, leading to slow computation and error accumulation effects [10]. Moreover, the self-attention mechanism can exhibit disorder, resulting in the loss of critical temporal information, even with the embedding of location and time data. Variations in sampling rates and sensor counts lead to significant differences and redundancy in multivariate time series data across scenarios. The cross-attention mechanism between the encoder and decoder often captures redundant information, further complicating the modeling process [25]. These factors collectively hinder the prediction performance of models on long sequences of spatio-temporal graph problems.

To address the above problems, we propose a long-range learnable graph Transformer based traffic flow prediction model, LL-Gformer. To address the first challenge on effective spatial-temporal construction, we propose a novel learning-based approach for constructing spatio-temporal graphs. For each recorded traffic flow information from sensors, we calculate the time similarity between different nodes using dynamic time warping and use this information to construct a time graph. Then, by combining the time graph with the spatial road network graph, we learn to construct both spatio-temporal correlated graphs and a global spatio-temporal graph that simultaneously incorporates time and space information, enabling accurate traffic flow prediction. To address the second challenge on limited exploitation of historical data, we incorporate traffic flow data from the same time period over the past week and the traffic data from the previous hour of the current day into the learning process for predicting future traffic conditions. For the third challenge on the limitations of model architecture, we introduce a new model. In the decoder, we incorporate a distillation mechanism that assigns higher weights to dominant attention features, effectively capturing long-range dependencies between long sequences of inputs. We replace the decoder with a fully connected mapping to directly learn the mapping between the features learned by the encoder and the predicted sequence. This approach avoids the problem of the Transformer decoder overly capturing redundant information in time series data, which can negatively impact predictive performance, and it reduces computational complexity. Additionally, the use of the global spatio-temporal graph significantly increases computation demands, so we propose two optimization strategies to reduce complexity. In summary, the contributions of this paper are as follows:

- We identify the importance of historical information and long-range spatio-temporal relationships in traffic prediction tasks, pointing out the the shortcomings of graph construction and model structure in existing methods.
- We propose a novel graph embedding method and encoding technique to learn the data representation for each sensor, and we design an efficient model to capture long-range dependencies in long sequence inputs while learning the mapping between features and the prediction sequence. Additionally, two optimization strategies are introduced to enhance the model's efficiency.
- Through experimental analysis, the proposed model can achieve advanced performance on four benchmark datasets.

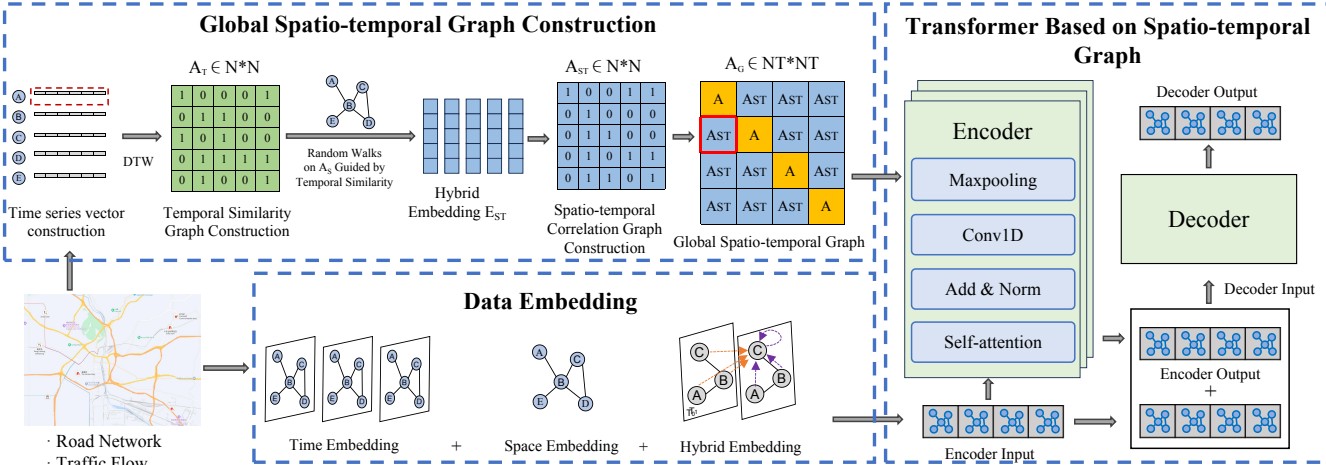

**Figure 2: The overall structure of LLGformer.**

## 2 Preliminaries

### 2.1 Notations and Definitions

*Definition 1 (Road Network)*: The input of the traffic flow forecasting is a network of roads. Road network can be represented as a weighted graph $\mathcal{G} = (\mathcal{V}, \mathcal{E}, A)$, where $\mathcal{V} = v_1, v_2, \cdots, v_N\}$ is the set of nodes, including $|\mathcal{V}| = N$, $N$ is the number of nodes in the graph. $\mathcal{E}$ is the set of edges, and $A \in \mathbb{R}^{N \times N}$ is the adjacency matrix of graph which stores the distances between sensors in the road network $\mathcal{G}$. $\mathcal{A}_{i,j} > 0$ if there exists an edge between node $i$ and node $j$, that is, $(i, j) \in \mathcal{E}$, otherwise 0.

*Definition 2 (Traffic Flow Tensor)*: We use $X^t \in \mathbb{R}^{N \times d}$ to denote the traffic flow of $N$ nodes in the road network observed at timestamp $t$, where $d$ is the number of features. We use $X = (X^1, X^2, \cdots, X^T) \in \mathbb{R}^{T \times N \times d}$ to denote the traffic flow tensor of all nodes over the total $T$ timestamps.

### 2.2 Problem Formalization

Traffic flow prediction aims to predict the traffic flow of a traffic system in a future period of time based on historical observations. We formalize the prediction as learning a traffic prediction model $f$, which is based on the road network $\mathcal{G}$ and predicts the traffic conditions for the future $T'$ timestamps based on the given $T$ historical timestamps of traffic conditions:

$$[\mathbf{X}^{(t-T):t}, \mathcal{G}] \xrightarrow{f} \mathbf{X}^{(t+1):(t+T')}. \tag{1}$$

## 3 Methodology

In this section, we first detail the method for integrating historical traffic information to construct a learnable global spatio-temporal graph. Next, we present the model architecture, which comprises encoding, encoder, and decoder components, effectively unifying spatio-temporal information. Finally, we introduce two optimization strategies to further reduce computational complexity and enhance model efficiency. The overall architecture of the proposed model is illustrated in Figure 2.

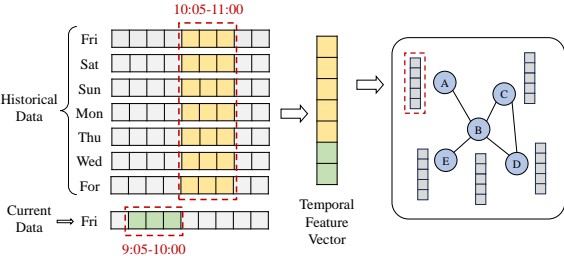

**Figure 3: For each node, we use the traffic flow data from the same time period in the past week as historical data (yellow in the figure) and the previous hour's traffic flow data as the current data (green in the figure). These historical and current data are combined to form the node's temporal feature vector.**

### 3.1 Global Spatio-temporal Graph Construction

The traffic data exhibits clear periodic patterns, with similar flow variations occurring at the same time each day across different dates. However, existing models typically only consider recent traffic data (e.g., from the past hour) when making predictions. To better capture these periodic trends, we incorporate historical traffic data from the same time slots of the previous week for each node. For instance, to predict traffic from 10:05 to 11:00 on a Friday, we also include data from the same time period over the past seven days (as shown in Figure 3).

**Temporal similarity graph**. For each sensor node, we create a time feature vector that incorporates both current and historical data. The time feature vectors for $N$ nodes are represented as $X_{\text{time}} = \{X_{\text{time}}^1, X_{\text{time}}^2, \ldots, X_{\text{time}}^N\} \in \mathbb{R}^{N \times T_{hc} \times d}$, where $T_{hc}$ is the total number of timestamps in both the historical and current data, and the $j$-th element of $X_{\text{time}}^i$ represents the traffic flow state of the $i$-th sensor at the $j$-th timestamp. The distance between two time feature vectors measures the temporal similarity between corresponding sensors. We compute this similarity using the Dynamic

Time Warping (DTW) algorithm [6]. For each node, we select the $k$ most similar nodes as neighbors and construct a time-related graph $A_T$ by assigning weights of 1 or 0 between nodes and their neighbors, as follows:

$$A_T[i,j] = \begin{cases} 1, & \text{if } v_j \text{ is a } k \text{ nearest neighbor of } v_i, \\ 0, & \text{otherwise.} \end{cases} \quad (2)$$

In the graph $A_T$, an edge between two nodes indicates similar urban functions and corresponding traffic patterns. Thus, the temporal similarity graph effectively captures the temporal correlations between different nodes.

**Spatio-temporal Correlation Graph**. The temporal similarity graph $A_T$ captures temporal correlations between nodes, while the road network graph $\mathcal{G}$ reflects spatial correlations. Our goal is to integrate these two graphs to construct the spatio-temporal correlation graph $A_{ST}$, which effectively combines both temporal and spatial information to capture the spatio-temporal dependencies.

We adopt a random walk strategy to integrate information from the temporal similarity graph and road network graph for node embedding. Inspired by FOGS [29], we propose a sampling approach that enables random walks on the road network to incorporate temporal correlations, addressing the limitations of traditional methods that rely solely on topological structures for neighbor discovery.

Let $\mathcal{G} = (N, \mathcal{E})$ represent the road network and $A_T = (N, \mathcal{E}_{\text{time}})$ represent the temporal correlation graph. Suppose a random walk starts from sensor $N_0$ and is currently at sensor $N_j$, with the path denoted as $\tau_j = \langle v_0, \ldots, v_{j-1}, v_j \rangle$, and the next sensor to visit is $v_{j+1}$. At this point, we incorporate temporal correlations: if there is an edge between nodes $v_0$ and $v_{j+1}$ in $A_T$, it indicates strong temporal correlation, increasing the probability of visiting $v_{j+1}$. Conversely, the absence of an edge suggests weaker temporal correlation, reducing the visit probability. This sampling strategy can be expressed as:

$$P(v_{j+1}|\tau_j) \propto \begin{cases} \frac{1}{p}, & \text{if } d = 0 \text{ and } A_T[v_0, v_{j+1}] = 1, \\ 1, & \text{if } d = 1 \text{ and } A_T[v_0, v_{j+1}] = 1, \\ \frac{1}{q}, & \text{if } d = 2 \text{ and } A_T[v_0, v_{j+1}] = 1, \\ 0, & \text{otherwise.} \end{cases} \quad (3)$$

where $d$ represents the shortest path distance between $v_{j+1}$ and $v_{j-1}$ in the road network $R$ within $\tau_j$, $p$ and $q$ are preset non-negative constants. The next sensor $v_{j+1}$ has three choices: to return to sensor $v_{j-1}$, the distance between sensor $v_{j+1}$ and sensor $v_{j-1}$ remains unchanged, or the distance from sensor $v_{j+1}$ to sensor $v_{j-1}$ increases by one. Therefore, the value of $d$ is 0, 1, or 2.

As a result, we can obtain the spatio-temporal mixed embedding $E_{ST}$ for all nodes, which integrates both geographical information from the road network and temporal correlations from the temporal similarity graph, effectively capturing spatio-temporal dependencies.Using the learned node embeddings $E_{ST}$, we construct a learning-based spatio-temporal correlation graph $A_{ST}$. We calculate the cosine similarity between the embeddings of each pair of nodes and select the $K$ most similar nodes as neighbors. For each node pair, $A_{ST}(i, j) = 1$ if they are neighbors, and 0 otherwise. In this graph, an edge between two nodes indicates both similar traffic patterns and spatial adjacency, effectively unifying temporal and spatial correlations within a single graph.

**Global Spatio-temporal Graph**. To capture long-range spatio-temporal dependencies in traffic data, we construct a global spatio-temporal graph. This graph enables free interaction between nodes across different locations and timestamps during the learning process, facilitating unified spatial and temporal aggregation,as follows:

$$A_G = \begin{bmatrix} A & A_{ST} & A_{ST} & \ldots & A_{ST} \\ A_{ST} & A & A_{ST} & \ldots & A_{ST} \\ A_{ST} & A_{ST} & A & \ldots & A_{ST} \\ \vdots & \vdots & \vdots & \ddots & \vdots \\ A_{ST} & A_{ST} & A_{ST} & \ldots & A \end{bmatrix} \in \mathbb{R}^{NT \times NT}. \quad (4)$$

As illustrated in Eq.(4), the global spatio-temporal graph $A_G$ consists of $T \times T$ submatrices, each of size $N \times N$. The diagonal submatrices represent the adjacency matrices $A$ of the road network $\mathcal{G}$, indicating that at a specific timestamp $t$, each node interacts with the features of other nodes without complex temporal relationships across different time slices. The off-diagonal submatrices $A_G[ti, tj]$ correspond to the adjacency matrices of the learned spatio-temporal correlation graph $A_{ST}$, capturing the interactions of each node's features between timestamps $t_i$ and $t_j$.

## 3.2 Data Embedding for Spatio-temporal Graph

The data embedding layer transforms raw input data into a high-dimensional representation. Recognizing the close interconnection between spatial and temporal information in traffic data, we developed a method that effectively integrates spatio-temporal characteristics into the model for enhanced processing.

**Spatio-temporal Hybrid Embedding**. Hybrid embedding captures the relevance of temporal and spatial information simultaneously. The sequences learned through the random walk strategy incorporate geographical data from the road network and temporal correlations from the temporal similarity graph, enhancing the representation of spatio-temporal relationships. Thus, $E_{ST}$ is used for spatio-temporal hybrid encoding. Historical data provides valuable periodic information, while current data is essential for accurate traffic prediction. To balance the contributions of historical and current information in mixed encoding, we introduce a gating module that fuses hybrid coding $\mathcal{X}_h$ (based on historical data) and $\mathcal{X}_c$ (focused on current information). The formulation is as follows:

$$E_{hybrid} = g(\Theta_1 * \mathcal{X}_h + a) \odot \sigma(\Theta_2 * \mathcal{X}_c + b). \quad (5)$$

where $E_{hybrid} \in \mathbb{R}^{N \times T \times D}$, $\Theta_1$, $\Theta_2$, $a$ and $b$ are model parameters, $\odot$ is the Hadamard product, $g(\cdot)$ is tanh function and $\sigma(\cdot)$ is sigmoid function.

**Time Embedding**. Time embedding involves the positional encoding of time series information.Here, we use a fixed encoding method that associates each sequence element with a point on a sine or cosine curve. The positional embedding is generated by combining sine and cosine functions of varying frequencies:

$$\begin{aligned} E_{(time,2i)} &= \sin(time/10000^{2i/D}) \\ E_{(time,2i+1)} &= \cos(time/10000^{2i/D}) \end{aligned} \quad (6)$$

where $E_{time} \in \mathbb{R}^{T \times D}$, $D$ is the dimension of the hidden layers, $time$ is the time position, and $T$ is the number of timestamps.

**Space Embedding**. Spatial embedding is the encoding of spatial information in the structure of a road network. First, information

is extracted based on the road network graph and the temporal sequence similarity of nodes to compute a topological graph. This topological graph represents the spatial position information between nodes, which does not change with the variation of time slices. Then, through matrix factorization, a normalized graph Laplacian matrix [5] is obtained, and its corresponding eigenvectors are used as the space embedding of the nodes, denoted as $E_{space}$. The Laplacian matrix and eigen decomposition can be represented as:

$$\Delta = I - D^{-1/2}AD^{-1/2} = U^T \Lambda U, \tag{7}$$

where $U \in \mathbb{R}^{N \times D}$ is the eigenvector corresponding to the Laplacian matrix, denoted as $E_{space}$. Here, $A$ is the adjacency matrix of the road network, $D$ is the degree matrix, $I$ is the identity matrix, and $\Lambda$ represents the eigenvalues of the Laplacian matrix.

The output $X_{input}$ of the data embedding layer is obtained by summing the three different forms of embedding vectors.

$$X_{input} = E_{time} + E_{space} + E_{hybrid}, \tag{8}$$

where $X_{input} \in \mathbb{R}^{B \times N \times T \times D}$, and $B$ is the number of samples selected in each training batchsize. $X_{input}$ will be used as input to the encoder-decoder structure below.

### 3.3 Encoder for Spatial-Temporal Graph

Existing spatio-temporal graph models typically split encoders into two parts: one for temporal information, and another for spatial information. Alternatively, separate modules may extract temporal and spatial features for fusion in a multi-head self-attention module. However, Existing methods often struggle to learn dynamic spatio-temporal dependencies in traffic data. Here, we propose a novel encoder structure based on the self-attention mechanism to accurately capture complex spatio-temporal correlations.

The encoder consists of $l$ layers of the same structure. In each layer, the attention mechanism operates in a fully connected manner. For the global information in the traffic graph, the $Q$, $K$ and $V$ matrices for different attention heads are first obtained using the global spatio-temporal graph $A_G$ constructed previously:

$$Q_i = A_G W_i^Q, K_i = A_G W_i^K, V_i = A_G W_i^V \tag{9}$$

where $W_i^Q$, $W_i^K$ and $W_i^V \in \mathbb{R}^{NT \times d_i}$. $d_i$ is the dimension of the $Q$ and $K$ matrices. Subsequently, the self-attention operation is used to model the interaction between nodes and obtain the attention score of the $i$-th attention head:

$$A_i = \left(\frac{Q_i K_i^\top}{\sqrt{d_i}}\right)V_i, \tag{10}$$

$A_i$ represents the dependencies among all nodes across all time slices, capturing global spatio-temporal correlations. The output matrix of the attention layer is obtained by multiplying the attention scores from different heads with $V$.

$$\text{Att}(Q_i, K_i, V_i) = \text{softmax}(A_i)V_i. \tag{11}$$

Different attention heads extract varying features, which are distinct due to different subspace representations of the same queries, keys, and values. For $h$ different attention heads, their outputs are concatenated to form a multi-head attention block:

$$\text{Multi-Head Att} = \text{Concat}(Att_1, Att_2, \ldots, Att_h)W^O, \tag{12}$$

where $W^O$ is a learnable projection matrix.

Following each attention layer, a residual connection [14] is employed to alleviate the vanishing gradient problem, while layer normalization [2] accelerates convergence. The encoder's feature representation is derived from the fully connected self-attention mechanism and concatenated with the encoder's input $X_{input}$ to form $X_i$. A distillation operation [52] is then applied to $X_i$, involving one-dimensional convolution followed by max pooling. This pooling operation reduces the input sequence length by half at each layer, decreasing the output feature dimensions and generating a concatenated attention feature map for the next layer. The distillation process from layer $i$ to $i+1$ is as follows:

$$X_{i+1} = \text{MaxPooling}(\text{ELU}(\text{1D-Conv}([X_i]))) \tag{13}$$

where $[\cdot]$ is the operations in the fully connected self-attention layer, ELU represents activation function. Through the distilling operation, features with dominant attention are given higher weights, enabling more effective processing of longer input sequences.

### 3.4 Decoder for Spatial-temporal Graph

The classic Transformer decoder generates sequences autoregressively by iteratively shifting the input right, which is slow and prone to cumulative errors, making it unsuitable for long-sequence traffic prediction. In contrast, the Trafformer utilizes a generative decoder with zero-padding equal to the encoder's length, allowing all predicted values to be computed in a single forward pass, significantly improving long-term prediction performance. However, it still struggles with the loss of temporal information and excessive redundancy in traffic data.

In this paper, we replace the decoder with a simple linear layer. The encoder employs a self-attention mechanism to capture temporal and spatial dependencies in traffic data, while the linear layer directly maps the encoder's learned features to the predicted sequence. The computation process is as follows:

$$\mathbf{X}_{de} = \text{Linear}(\mathbf{X}_{input} + \mathbf{X}_{en}). \tag{14}$$

where $X_{en}$ is the output of the encoder. The linear layer achieves a more concise and efficient calculation, which is more suitable for the spatio-temporal graph prediction task of long sequences.

### 3.5 Strategies to Reduce Complexity

The complexity of the classical Transformer self-attention module is $O(T^2)$. We apply the global spatio-temporal graph to the Transformer model, which makes the complexity $O(N^2 \times T^2)$. In order to reduce the computational complexity, two optimization strategies are proposed.

**Sparse Mask Matrix Strategy**. Self Attention calculates the relevance between every pair of vectors in a sequence, resulting in a correlation matrix with a computational complexity of $O(N^2)$ for an input sequence of length $n$. In this paper, we construct a global spatio-temporal graph of size $NT \times NT$ to capture the interdependence of all nodes across time slices. While this approach effectively learns spatio-temporal correlations in traffic data, it also incurs significant computational costs. To conserve memory and enhance computation speed, we propose reducing the calculation of interdependencies by considering only a subset of elements in the sequence, thereby decreasing the computational load from a sparse matrix perspective.

In constructing the spatio-temporal correlation graph, we select the $k$ most similar nodes as neighbors for each node, controlling the graph's sparsity by adjusting $k$. We create a new spatio-temporal correlation matrix with an optimal value $k'$, which serves as the mask matrix $W_{mask}$. The attention matrix is then masked using the following formula:

$$A_{\text{spare}} = W_{\text{mask}} \odot (Q \times K), \tag{15}$$

where $\odot$ is Hadamard product.

After applying the mask to the fully connected self-attention matrix, we obtain a sparse attention matrix $A_{\text{spare}}$, significantly reducing the computational load. This adjustment lowers the computational complexity to $O(E \times T^2)$.

**Memsizer Optimization Strategy**. In Transformer-based models, the computation of attention involves calculating softmax$(QK^T)$, followed by matrix multiplication with $V$, which restricts gains in computational efficiency. In this paper, we adopt the optimization strategy from Memsizer [2] to enhance the efficiency of the attention component.

In Transformer, the self-attention mechanism uses the original vector $X_s$ and the target vector $X_t$ as inputs, generating $K$ and $V$ from $X_s$ and $Q$ from $X_t$. We propose a new mechanism to replace the self-attention module, implementing recursive reasoning computations. Here, the attention component is computed as $\alpha = f(QK^T)$ and $X_{out} = \alpha V$, where $K$ and $V$ are considered pointwise projections of the original vector $X_s$. The three matrices can be represented in the following new form:

$$\begin{aligned} Q &= X^t, \quad K = \Phi, \\ V &= \text{LN}(W_l(X_s)^T)\text{LN}(X_s W_r). \end{aligned} \tag{16}$$

where, $K \in \mathbb{R}^{k \times D}$, LN denotes layer normalization, $W_l \in \mathbb{R}^{D \times k}$, and $W_r \in \mathbb{R}^{D \times D}$. $K$ is a trainable matrix shared across instances, rather than derived from a linear transformation of the input. This matrix is significantly smaller than the $K$ generated by the original Transformer, reducing the computational complexity of the attention mechanism. For the value, the input $X_s$ is first weighted using $W_l$ and then normalized, followed by a linear transformation and another normalization to facilitate interaction at each matrix position. In the multi-head attention mechanism, $K$ is independent for each head while $V$ is shared, enabling recursive cyclic computations across heads. Finally, average pooling is applied to the output of each head, resulting in the following output representation:

$$X_{out} = 1/r \cdot \sum_{i=1}^{r} X_{out}^i \tag{17}$$

Consequently, the computational complexity is reduced to $O(NTk)$, where the value of $k$ is much smaller than $N$ and $T$, effectively achieving linear complexity.

## 3.6 Loss Function

In this paper, we choose the Mean Square Error (MSE) as the loss function. The objective function is shown as follows:

$$L(\hat{\mathbf{X}}^{(t+1):(t+T)}; \Theta) = \frac{1}{TN} \sum_{i=1}^{i=T} \sum_{j=1}^{j=N} (\hat{\mathbf{X}}_j^{(t+i)} - \mathbf{X}_j^{(t+i)})^2, \tag{18}$$

where $\Theta$ is the model parameter.

## 4 Experiments

We first outline the experimental setup and compare the new LLGformer approach with state-of-the-art methods for traffic flow prediction. Additionally, we conduct analyses of computational efficiency, ablation studies, and parametric analysis to further assess the effectiveness of the proposed method.

## 4.1 Experimental Setup

**Datasets**.We evaluate the performance of the proposed model using four real-world datasets: PEMS04, PEMS08, METR-LA, and PEMS-BAY. The detailed description and statistical information of the datasets are provided in Appendix B.1.

**Baselines**. We compare LLGFormer with seven existing state-of-the-art models, including ARIMA[41], GraphWavenet[44], AGCRN[3], Trafformer[19], ASTGCN[12], STSGCN[34],and STFGNN[24]. Detailed descriptions of these models are provided in Appendix B.2.

**Implementation details**. For all baselines, we use the source codes released by their authors. For the proposed LLGformer, the number of encoder layers is set to 2, and the hidden layer dimension for each attention layer is set to 32. In the random walk strategy, the values of parameters $p$ and $q$ are both set to 1. When constructing the time-related graph $A_T$ and spatio-temporal-related graph $A_{ST}$, the value of $k$ is set to 12. Different batch sizes are set for experiments on different datasets. The batch size for METR-LA is set to 11, PEMS-BAY to 5, and PEMS04 and PEMS08 to 16. The model is trained using the Adam optimizer with a learning rate of 0.001. The datasets are sorted in ascending order of time and split into training (70%), validation (10%), and testing (20%) sets. We utilize three common metrics for traffic flow prediction: MAE, MAPE, and RMSE. The calculation formulas are provided in Appendix B.3.

## 4.2 Experimental Results and Analysis

To validate the model's effectiveness, traffic prediction experiments were conducted on four different datasets. For the METR-LM and PEMS-BAY datasets, we forecasted traffic conditions for the next 15, 30, and 60 minutes. For the PEMS04 and PEMS08 datasets, we predicted traffic conditions for the next hour. Lower values for the three evaluation metrics indicate better model performance.

The experimental results (Tables 1 and 2) indicate that LLGformer consistently outperforms all other models across datasets, demonstrating state-of-the-art capabilities in both long-term and short-term predictions. The classical ARIMA method, which relies solely on time-related data, struggles with non-stationary temporal relationships, resulting in poor predictive performance. Models like ASTGCN, STFGNN, and STSGCN leverage spatial information and outperform basic time series methods. However, they typically model spatial and temporal relationships separately, particularly ASTGCN, while STFGNN and STSGCN focus on local dependencies, leading to suboptimal outcomes. In short-term tasks like 15-minute forecasts, Graph Wavenet performs well but struggles with longer predictions due to limitations in stacking spatiotemporal layers and expanding the receptive field of 1D-CNN. Trafformer constructs a global spatiotemporal graph that enables direct interaction among all spatial positions across time slices, outperforming Graph Wavenet in long-range predictions. Despite some limitations, LLGformer effectively combines historical information to model

**Table 1: Performance Comparison of LLGformer and other Baseline Models in Traffic Prediction Task on METR-LA and PEMS-BAY Datasets. Lower Values Indicate Better Performance.**

| Dataset | Models | 15 min | | | 30 min | | | 60 min | | |
|---|---|---|---|---|---|---|---|---|---|---|
| | | MAE | RMSE | MAPE(%) | MAE | RMSE | MAPE(%) | MAE | RMSE | MAPE(%) |
| METR-LA | ARIMA | 3.99±0.12 | 8.21±0.16 | 9.60±0.10 | 5.15±0.22 | 10.45±0.25 | 12.70±0.10 | 6.90±0.00 | 13.23±0.32 | 17.40±0.15 |
| | AGCRN | 3.67±0.00 | 9.58±0.04 | 8.45±0.02 | 4.75±0.11 | 12.10±0.05 | 10.77±0.27 | 6.13±0.13 | 14.86±0.09 | 13.46±0.16 |
| | ASTGCN | 2.96±0.05 | 5.71±0.00 | 7.81±0.13 | 3.44±0.09 | 6.62±0.12 | 9.33±0.23 | 3.85±0.19 | 7.79±0.04 | 10.88±0.21 |
| | STFGNN | 3.21±0.04 | 6.52±0.02 | 8.14±0.09 | 3.51±1.13 | 6.62±0.09 | 9.77±1.10 | 3.86±0.09 | 7.65±0.15 | 10.89±0.13 |
| | STSGCN | 3.43±0.23 | 6.57±0.19 | 9.73±0.25 | 3.60±0.17 | 6.96±0.20 | 10.35±0.07 | 3.95±0.10 | 7.77±0.16 | 11.65±0.09 |
| | Graph Wavenet | 2.69±0.00 | 5.15±0.04 | 6.93±0.02 | 3.07±0.10 | 6.22±0.05 | 8.37±0.05 | 3.53±0.11 | 7.37±0.08 | 10.01±0.10 |
| | Trafformer | 2.78±0.05 | 5.35±0.02 | 7.32±0.04 | 3.05±0.08 | 6.18±0.05 | 8.67±0.10 | 3.41±0.10 | 7.17±0.13 | 9.96±0.11 |
| | **LLGformer** | **2.42±0.00** | **4.91±0.03** | **6.64±0.02** | **3.01±0.11** | **6.02±0.09** | **8.14±0.13** | **3.15±0.00** | **6.89±0.03** | **9.38±0.01** |
| PEMS-BAY | ARIMA | 1.82±0.08 | 3.30±0.11 | 3.50±0.06 | 2.33±0.23 | 4.76±0.19 | 5.40±0.15 | 3.38±0.32 | 6.51±0.28 | 8.34±0.19 |
| | AGCRN | 2.14±0.11 | 4.85±0.09 | 4.65±0.13 | 1.76±0.20 | 3.97±0.17 | 3.82±0.17 | 1.39±0.09 | 2.98±0.11 | 4.20±0.14 |
| | ASTGCN | 1.92±0.03 | 3.98±0.03 | 4.27±0.02 | 1.82±0.12 | 3.95±0.15 | 4.16±0.20 | 2.04±0.23 | 4.65±0.30 | 4.22±0.26 |
| | STFGNN | 2.25±0.17 | 4.35±0.20 | 5.41±0.19 | 2.42±0.31 | 4.25±0.26 | 5.88±0.15 | 2.54±0.21 | 4.89±0.24 | 5.71±0.19 |
| | STSGCN | 2.54±0.18 | 4.47±0.23 | 5.88±0.20 | 2.61±0.15 | 4.93±0.15 | 6.03±0.21 | 2.71±0.18 | 5.28±0.25 | 6.39±0.22 |
| | Graph Wavenet | 1.32±0.05 | 2.74±0.03 | 2.73±0.03 | 1.63±0.17 | 3.70±0.10 | 3.67±0.15 | 1.95±0.09 | 4.52±0.17 | 4.63±0.14 |
| | Trafformer | 1.88±0.20 | 4.38±0.19 | 4.59±0.24 | 1.61±0.06 | 3.74±0.03 | 3.82±0.03 | 1.31±0.11 | 2.83±0.14 | 2.92±0.13 |
| | **LLGformer** | **1.16±0.07** | **2.68±0.10** | **2.54±0.05** | **1.59±0.13** | **3.26±0.09** | **3.42±0.09** | **1.22±0.02** | **2.61±0.00** | **2.77±0.06** |

**Table 2: Performance Comparison of LLGformer and other Baseline Models in Traffic Prediction Task on PEMS04 and PEMS08 Datasets. Lower Values Indicate Better Performance.**

| Models | PEMS04 | | | PEMS08 | | |
|---|---|---|---|---|---|---|
| | MAE | RMSE | MAPE(%) | MAE | RMSE | MAPE(%) |
| ARIMA | 23.71±0.11 | 36.88±0.17 | 17.61±0.14 | 19.02±0.09 | 29.88±0.14 | 13.35±0.14 |
| AGCRN | 19.74±0.09 | 32.01±0.03 | 12.98±0.08 | 15.92±0.14 | 25.31±0.08 | 10.30±0.11 |
| ASTGCN | 22.90±0.32 | 32.59±0.27 | 16.75±0.22 | 18.72±0.17 | 28.99±0.20 | 12.53±0.17 |
| STFGNN | 19.68±0.03 | 31.85±0.09 | 13.07±0.04 | 15.87±0.13 | 24.98±0.15 | 10.41±0.15 |
| STSGCN | 21.19±0.07 | 33.65±0.04 | 13.90±0.05 | 17.13±0.12 | 26.80±0.12 | 10.96±0.07 |
| Graph Wavenet | 19.91±0.21 | 31.06±0.24 | 13.62±0.18 | 15.57±0.11 | 24.32±0.09 | 10.32±0.12 |
| Trafformer | 19.26±0.21 | 30.67±0.25 | 12.96±0.15 | 15.27±0.09 | 24.33±0.15 | 10.19±0.20 |
| **LLGformer** | **19.12±0.09** | **30.59±0.09** | **12.88±0.14** | **15.16±0.06** | **24.21±0.15** | **10.08±0.08** |

global spatiotemporal correlations across various positions and time slices, optimizing its structure for superior performance in long time series predictions. In long-range prediction tasks on the METR-LA dataset, LLGFormer's MAPE outperforms ARIMA, AGCRN, AST-GCN, STFGNN, STSGCN, Graph Wavenet, and Trafformer by 8.02%, 4.08%, 1.5%, 1.51%, 2.27%, 0.63%, and 0.58%, respectively. These experiments affirm that the proposed model exhibits excellent predictive performance, with a more pronounced improvement in accuracy for long-range predictions.

### 4.3 Computation Efficiency Analysis

In order to visualize the time complexity of the proposed LLGformer model in this paper and validate the practical effects of different optimization strategies, we compared the original LLGformer model with variants using optimization strategies and contrasted them with the traditional Transformer model. The dataset used is METR-LA, and evaluation metrics is MAE, with specific results presented in Table 3.Here, $M$ represents the Memsizer Optimization Strategy, $S$ represents the Sparse Mask Matrix Strategy, $ST$ represents using the spatio-temporal correlation matrix as the mask matrix, and $A$ indicates directly using the adjacency matrix as the mask matrix. When employing the Memsizer Optimization Strategy, the theoretical computational complexity is $O(NTk)$, achieving linear complexity and the highest computational efficiency among all variants. However, due to local interactions between tokens during the computation, there is a slight impact on experimental performance. The Sparse Mask Matrix Strategy theoretically has a computational complexity of $O(ET^2)$, providing less remarkable improvement in computational efficiency. Still, when using the spatio-temporal correlation graph as the sparse mask matrix, it has a relatively minor impact on the model's performance.

### 4.4 Ablation Study

To investigate the effectiveness of different modules in LLGformer, we designed five model variants for ablation experiments. First, to assess the learned spatio-temporal fusion graph, two variants were created: one using a traditional attention mechanism, denoted as "$-AGST$", and another that concatenates the road network's adjacency matrix to form a global spatio-temporal graph, denoted as "$+A$". Next, we removed the hybrid encoding, relying solely on conventional temporal and spatial encoding, referred to as "$-Ehybrid$". Another variant maintained the decoder's iterative output structure

**Table 3: The time consumpiton on METR-LA dataset**

| Models | MAE | Training | inference |
|--------|-----|----------|-----------|
| LLGformer | 3.15 | 877.43 | 58.96 |
| LLGformer-M | 4.09 | 310.24 | 13.58 |
| LLGformer-S-ST | 3.64 | 586.97 | 26.88 |
| LLGformer-S-A | 4.16 | 658.73 | 29.82 |

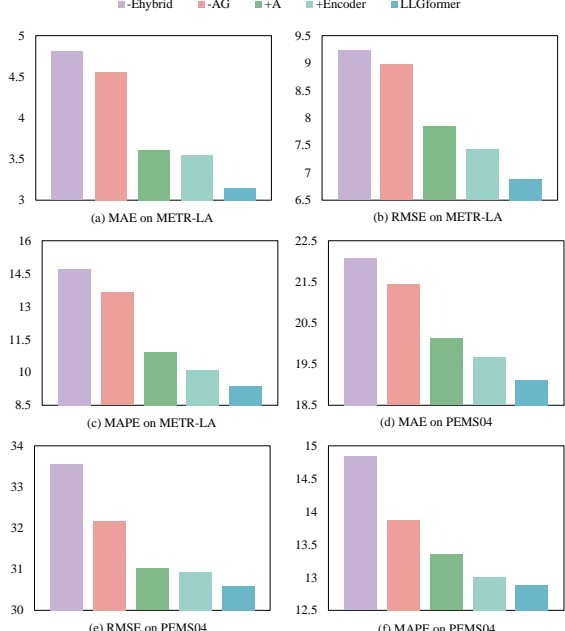

**Figure 4: Ablation Study on METR-LA and PeMS04.**

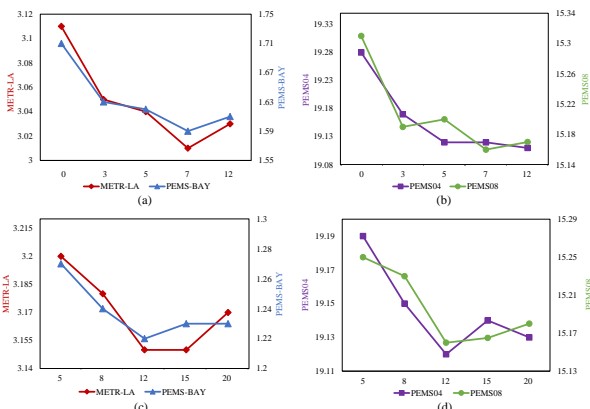

**Figure 5: Parameter analysis.(a) and (b) show the impact of the days of historical data observation. (c) and (d) show the impact of the k-order nearest neighbors.**

without using a linear layer, denoted as "*+Encoder*". The experiments were conducted on the METR-LA and PEMS04 datasets, evaluating performance with MAE, RMSE, and MAPE metrics. The results are presented in Fig. 4.

The experimental results demonstrate that the proposed LLGformer model outperforms variants that omit certain components. Specifically, excluding $A_{GST}$ prevents direct interactions between different spatial locations across various time slices, while constructing a global spatio-temporal graph solely with the adjacency matrix $A$ fails to incorporate temporal and historical information. These findings highlight the importance of the learnable spatio-temporal fusion graph combined with historical data. Additionally, removing hybrid encoding hinders the model's ability to capture spatio-temporal dependencies between nodes. The "*+Encoder*" variant further exacerbates overfitting in the Transformer-style model for long time series predictions, negatively impacting accuracy.

### 4.5 Parametric Analysis

**Days of Historical Data Observation:** To construct the temporal similarity graph and implement spatio-temporal hybrid encoding for data embedding, we analyzed the impact of different historical

data observation periods on model performance. Experiments were conducted with variants using historical traffic data observed for 0, 3, 7, 10, and 14 days, evaluating performance through the MAE for 30-minute predictions across four datasets. Results are shown in Fig. 5(a) and Fig. 5(b). As indicated, the variant using 0 days of historical data performed the worst, highlighting the importance of historical data in traffic prediction. The best performance was achieved with data from the past 7 days. Using fewer days may hinder the capture of periodic patterns or sudden traffic changes, while too many days can introduce noise and increase computational complexity. Thus, we opted to use historical data from the past 7 days.

***k*-order Nearest Neighbors:** When constructing spatio-temporal correlation graphs, the value of $k$ determines the number of edges. To assess the impact of $k$-order nearest neighbors on model performance, we conducted experiments with different variants using 5, 8, 12, 15, and 20 neighbors. The evaluation metric was the MAE for 60-minute predictions across four datasets, with results shown in Fig. 5(c) and Fig. 5(d). As $k$ increases, both the spatio-temporal correlation graph and the global spatio-temporal graph gain more edges, which may enhance information and improve accuracy. However, excessive increases can lead to irrelevant information and added computational burdens. Thus, we set $k = 12$ in the final model.

## 5 Conclution

This paper presents a novel LLGformer model based on learnable long-range graphs for spatio-temporal traffic prediction. A new graph embedding method and encoding scheme are designed to learn the data representation of each sensor, enabling the model to capture periodic patterns in traffic data through historical information. This representation better reflects the temporal and spatial correlations among sensors. A simple yet efficient method is proposed to capture long-range dependencies between input sequences and learn the mapping between features and prediction sequences. Additionally, to reduce the computational complexity of the model, two variants of the LLGformer model are introduced to improve training efficiency. The experiments on four real datasets confirm the effectiveness of the LLGformer model in traffic prediction tasks.

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

# A  Related Work

Traffic flow prediction, characterized by the complexity of its spatio-temporal relationships and the high nonlinearity of traffic data, has increasingly become a significant issue of common interest in the academic community. The primary focus of research on traffic flow prediction lies in the learning of spatial information, temporal information, and their interdependencies.

## A.1  Time Series Forcasting

In the field of traffic flow forecasting, learning time series information can be effectively managed using time series forecasting models. Historically, due to the significance of time series prediction problems, several classical models based on statistics or machine learning have been extensively studied. For instance, the Autoregressive Integrated Moving Average (ARIMA) [41] model transforms non-stationary data into stationary data for regression analysis. Support Vector Regression (SVR) [42] employs nonlinear mapping to project data into a high-dimensional feature space. However, traditional machine learning methods exhibit limitations in processing highly nonlinear traffic data and capturing spatio-temporal dependencies. Deep learning approaches have offered new perspectives for addressing traffic prediction challenges. The Recurrent Neural Network (RNN) model, for example, models time dependencies in time series. The TT-RNN [50] encodes high-order non-Markovian dynamics and state interactions, employing a tensor recursive architecture to learn nonlinear dynamics and address long-term time dependencies and high-order correlation issues. LST-Net [21] utilizes CNNs combined with RNNs to extract short-term local dependency patterns between variables and learn long-term time series patterns. The Deep State Space model [28] models the relationship between two consecutive hidden states and learns a global shared mapping relationship through covariance related to each time series. RNN-BLSTM [1] employs a traffic similarity strategy to cluster all BSs and uses the aggregated traffic pattern for multivariate spatio-temporal traffic prediction. Due to gradient vanishing, RNNs struggle to capture periodic temporal correlations and are computationally expensive. In contrast, Temporal Convolutional Networks (TCN) [4] with parallel 1D-CNN architecture offer more efficient operations. DeepSTN [26] uses the ConvPlus structure, combining PoI distribution and time factors to model long-range spatial dependencies between people in different regions, and expresses the influence of location attributes. DeepGLO [32] combines the matrix factorization model with the local deep temporal model through a data-driven attention mechanism to achieve global thinking and local action. GLU [8] proposed a CNN-based gating mechanism that can analyze inputs hierarchically to enhance temporal learning capabilities. The attention mechanism has also been applied to RNNs to help capture the long-term correlation of time series. For example, TPA-LSTM [33] introduces an attention mechanism to enhance the model's focus on different timestamps and better capture important information in time series. The attention-based Transformer [37] is a highly efficient method with excellent performance in capturing long-range temporal relationships between different timestamps. Based on the Transformer, Informer [52] adopts the ProbSpare self-attention mechanism, improving model efficiency and achieving excellent results in dealing with very long sequence prediction. Autoformer [43] proposed the Auto-Correlation Mechanism and applied it to the deep decomposition architecture to realize the trend and periodic decomposition of time information and improve the utilization efficiency of sequence information. PDFormer designs a delay-aware Feature Transformation module and applies it to the spatio-temporal self-attention module, which can explicitly model the time Delay of spatial information propagation.

## A.2  Spatio-Temporal Graphs

For spatial information in traffic flow prediction tasks, spatio-temporal graphs are often used for learning. Graph Neural Networks (GNN) [31] are effective tools for handling spatial dependencies in non-Euclidean spaces, demonstrating excellent performance in extracting spatial information and learning spatio-temporal graphs. STGCN [49] uses convolutional networks combined with residual links to construct spatio-temporal convolution blocks, extracting multi-scale spatio-temporal correlations of traffic networks. USTGCN [17] facilitates spatial and temporal aggregation through spectral graph convolution on spatio-temporal graphs, enabled by direct information propagation across nodes at different timestamps. In practical scenarios of traffic prediction, spatio-temporal dependencies continuously change over time, and the displayed graph structure may not reflect real dependencies. Using a fixed adjacency matrix fails to capture evolving spatio-temporal associations. Graph WaveNet [44] proposes an adaptive dependency matrix and uses different granular levels of dilated causal convolution layers for multi-scale information aggregation, demonstrating strong long-sequence prediction capabilities. Influenced by complex surrounding factors such as regional urban functions, lane numbers, traffic speed limits, and traffic control, traffic data does not entirely conform to geographical proximity, exhibiting uneven distributions in both spatial and temporal aspects. FOGS [29] uses gradient learning from traffic data changes to train trend models for prediction, avoiding overfitting and irregular distribution issues. Bi-STAT [7] designs a Dynamic Halting Module (DHM) cycle mechanism, dynamically resolving traffic prediction problems based on unique spatio-temporal complexities. The attention mechanism also plays a significant role in capturing spatio-temporal correlations. ST-DGN [51] constructs a hierarchical graph neural architecture to learn local and global spatial semantic information, utilizing multi-scale attention networks to capture multi-level temporal dynamics. LSGCN [16] proposes the COSATT graph attention network, combining it with GCN for spatial gating blocks and linear unit convolution, iteratively predicting future traffic flow.

# B  Details of Experimental Settings

## B.1  Datasets

We evaluate the performance of the proposed model using four real-world datasets: PEMS04, PEMS08, METR-LA, and PEMS-BAY. PEMS04 records continuous traffic flow data from 307 sensors for 59 consecutive days starting from January 1, 2018. PEMS08 records traffic flow data from 170 sensors for 62 consecutive days starting from July 1, 2016. Metro-la captures data from 207 sensors on the Los Angeles freeway from March 1, 2012, to June 30, 2012, covering four months of traffic information. PEMS-BAY collects traffic data

**Table 4: Data Description.**

| Data | #Nodes | #Edges | #Days | #Time Steps |
|---|---|---|---|---|
| METR-LA | 207 | 1515 | 119 | 34272 |
| PEMS-BAY | 325 | 2369 | 181 | 52116 |
| PEMS04 | 307 | 340 | 59 | 16992 |
| PEMS08 | 170 | 295 | 62 | 17856 |

from 325 loop detectors on the Los Angeles freeway for six months, from January 1, 2017, to May 31, 2017. During the experiment, the datasets are sorted in ascending order of time and split into training (70%), validation (10%), and testing (20%) sets. The same random splitting method is used for all methods. Table 4 provides detailed statistics for the datasets.

## B.2 Baselines

The detailed description of the comparison baseline is as follows:

(1) ARIMA [41]: Autoregressive Integrated Moving Average model, which transforms non-stationary data into stationary data and then establishes a model through regression.

(2) GraphWavenet [44]: Utilizes an adaptive adjacency matrix for encoding and employs expanding causal convolution layers at different granularity levels for graph convolution.

(3) Trafformer [19]: Applies a global spatio-temporal graph to the transformer model, using self-attention mechanisms to learn dependencies between nodes at different time slices and geographical locations.

(4) AGCRN [3]: Adaptive Graph Convolutional Recurrent Network, combines with gated neural units to learn patterns for specific nodes in traffic prediction.

(5) ASTGCN [12]: Attention-based spatio-temporal Graph Convolutional Network, designed with time attention and space attention mechanisms to capture temporal and spatial information separately.

(6) STSGCN [34]: The temporal graph and temporal connection graph are used to construct the STGC module, and the spatio-temporal dependencies are learned in the stacked spatio-temporal fusion graph neural layer.

(7) STFGNN [24]: Designs spatio-temporal convolution modules, utilizing graph convolutional networks to capture local spatio-temporal correlations.

## B.3 Evaluation Metrics

Three metrics are employed in the experiments: Mean Absolute Error (MAE), Mean Absolute Percentage Error (MAPE), and Root Mean Square Error (RMSE). The metrics are defined as:

$$MAE = \frac{1}{n} \sum_{i=1}^{n} |y_i - \hat{y}_i| \tag{19}$$

$$MAPE = \frac{1}{n} \sum_{i=1}^{n} \left| \frac{y_i - \hat{y}_i}{y_i} \right| \times 100\% \tag{20}$$

$$RMSE = \sqrt{\frac{1}{n} \sum_{i=1}^{n} (y_i - \hat{y}_i)^2} \tag{21}$$

where $n$ is the number of nodes, $y_i$ is the true value, and $\hat{y}_i$ is the predicted value by the model.