# OpenReview forum: "LLGformer: Learnable Long-range Graph Transformer for Traffic Flow Prediction"
_ACM.org/TheWebConf/2025/Conference — WWW 2025 Poster_

### Official Review · Reviewer_SsUk · 2024-11-23

**Novelty:** 4
**Technical Quality:** 6

**Review:**

This paper focuses on the well studied problem of predicting the delay of road segments in road networks (specifically highway networks). The authors present a new architecture for solving this problem. The novel components of the architecture are essentially:
1) (Re)introducing the historical delay at the given time of day (or week?) as a feature.
2) A random walk based approach to construct the spatio-temporal adjacency matrix.
3) Modified encoders and decoders.

The authors then proceed to show that their method beats the state of the art solutions on the traditional datasets of METR-LA and PEMS-BAY which essentially have delay observations of sensors on the highway networks of Los Angeles and San Francisco metro areas respectively.

As fast as criticisms are concerned, I believe the paper lacks a strong novelty component in the sense that the ideas introduced are by no means groundbreaking. One of them is in fact the oldest trick in the book (historical average). It also lacks any sort of theoretical justification of the proposed architectural decisions.

On the other hand, if the experimental evaluation is to be trusted -- and I don't see any reason against it -- the architecture does beat the state of the art in an important and extensively studied problem. As far as I can tell the authors did compare against the best known solution for the problem.

**Questions:**

Was there really no prior work that incorporated the historical cost? Or was it just omitted by the best net based architectures?

**Reviewer Confidence:**

3: The reviewer is confident but not certain that the evaluation is correct

**Scope:**

3: The work is somewhat relevant to the Web and to the track, and is of narrow interest to a sub-community

---

### Official Review · Reviewer_X77v · 2024-11-29

**Novelty:** 4
**Technical Quality:** 4

**Review:**

This paper proposes LLGformer, a spatio-temperal graph model to predict traffic flow in road networks. The graph consists of all time-space pairs as nodes and edges are constructed with multiple similarity metrics. Node embeddings include time, space and hybrid information, which are fed into an encoder-decoder structure. Authors provide experimental results on four datasets and show improved performance. They also provide two acceleration strategies, which sacrifice performance partly.

Pros:

* The long-range feature construction is intuitive and the random walk strategy for node embedding is interesting.
* Many baselines are evaluated and the performance of LLGformer leads in Table 1&2.
* Ablation studies and parameter analysis are included.
* The manuscript is overall clear and well-structured.

Cons:

* The novelty of LLGformer compared to Trafformer [19] is somewhat limited. The graph representation, space and time encoding, encoder & decoder structures and the optimization strategy are identical, and the manuscript does not mention this point. The incremental part only includes different edge construction of $A_{ST}$ and the Memsizer optimization strategy.
* My major concern is about the efficiency evaluation of LLGformer (and Trafformer). The two models constructs a much larger graph of size $NT \times NT$ compared to prior works, and it should be expected to improve the prediction accuracy. LLGformer, on top of Trafformer, introduces even more complex feature construction (fusing historical data). The training and inference time should be included in Table 1&2 for a fair comparison, as the optimized version of LLGformer-M (in Table 3) does not necessarily outperform most baselines.

**Questions:**

1. What is the length of time steps?
2. Page 2 says "Moreover, the self-attention mechanism can exhibit disorder, resulting in the loss of critical temporal information, even with the embedding of location and time data." Can you elaborate on this or support this claim with examples or evidence?
3. Increase the font size in Fig. 1,4,5 for readability.
4. Typo: Section 5 conclusion

**Reviewer Confidence:**

3: The reviewer is confident but not certain that the evaluation is correct

**Scope:**

3: The work is somewhat relevant to the Web and to the track, and is of narrow interest to a sub-community

---

### Official Review · Reviewer_71aP · 2024-12-02

**Novelty:** 5
**Technical Quality:** 6

**Review:**

This paper proposes the LLGformer model, which addresses the limitations of existing traffic flow prediction methods in spatiotemporal graph construction, historical data utilization, and model architecture design. By introducing a learnable spatiotemporal graph construction method and incorporating historical data, LLGformer effectively captures long-range spatiotemporal dependencies and periodic patterns, while optimizing the decoder structure to improve prediction efficiency. Experimental results demonstrate that LLGformer outperforms existing methods on multiple real-world datasets, showcasing its potential for application in intelligent transportation systems.

Pros:
1.The paper clearly outlines the limitations of existing traffic flow prediction methods, including issues with spatiotemporal graph construction, historical data utilization, and model architecture design. The authors provide an in-depth analysis of how these issues impact prediction accuracy and efficiency, and explicitly highlight the key areas for improvement, offering strong motivation for the development of the LLGformer model.
2.The paper proposes a learnable spatiotemporal graph construction method based on Dynamic Time Warping (DTW), which effectively captures long-range spatiotemporal dependencies. It also incorporates data from the same time period over the past week to enhance the model's ability to capture periodic variations. Furthermore, the optimized model architecture, incorporating a distillation mechanism and improvements to the decoder, increases prediction efficiency, while the proposed computational optimization strategies enhance the model's efficiency and scalability.
3.The paper demonstrates the effectiveness of the LLGformer model through extensive experiments on four real-world traffic datasets. The experimental results show that the model significantly outperforms existing methods across multiple datasets.
4.The paper is well-structured, with a clear organization of sections and logically coherent content. It is easy to follow and understand, which makes the paper highly readable and convincing.

**Questions:**

Cons:
1.The paper does not fully explain the reasons for selecting time alignment algorithms or similar time series alignment and similarity calculation methods. The time regularization algorithm is usually used to deal with the problem of time synchronization in time series data, but the paper does not explicitly explain why such problems exist in the task of traffic flow prediction, and why this algorithm is more suitable for this task than other similarity calculation methods. Therefore, this point requires more detailed explanation and argumentation from the author.
2.Although the paper mentions some related work, the literature review section may have overlooked some important methods for spatiotemporal sequence forecasting, especially those that are closely related to the ideas or techniques of the LLGformer model. For example, [1][2].
[1] Zezhi Shao, Zhao Zhang, Wei Wei, Fei Wang, Yongjun Xu, Xin Cao, Christian S. Jensen: Decoupled Dynamic Spatial-Temporal Graph Neural Network for Traffic Forecasting. Proc. VLDB Endow. 15(11): 2733-2746 (2022)
[2]Minhao Liu, Ailing Zeng, Muxi Chen, Zhijian Xu, Qiuxia Lai, Lingna Ma, Qiang Xu:SCINet: Time Series Modeling and Forecasting with Sample Convolution and Interaction. NeurIPS 2022

**Reviewer Confidence:**

4: The reviewer is certain that the evaluation is correct and very familiar with the relevant literature

**Scope:**

4: The work is relevant to the Web and to the track, and is of broad interest to the community

---

### Official Review · Reviewer_hVfH · 2024-12-02

**Novelty:** 5
**Technical Quality:** 4

**Review:**

The article deals with traffic prediction in a city, using a spatio-temporal model of a graph to learn patterns and predicts flow with a Transformer-based architecture.

The paper is well written, with rather good clarity. A few minor typos here and there (e.g, spaces between sentences, or "conclution").

The work follows a trend of (graph) machine-learning papers on traffic prediction, from CNNs and RNNs to GNNs. The originality of the paper lies in the model it develops: while a spatio-temporal model with attention is not new (e.g., it has been used for other tasks, such as general link prediction, see "cons" below), the model presented here *is new* and has some genuine originality. Moreover, it is the first on flow prediction, and achieve good performance, so it has some significance for the community.


Pros:
- original spatio-temporal model, used for the first time on this task
- good experimental performance

Cons:

- in terms of scope, the link between traffic prediction and "the Web" is thin. So, while the method and the study in this paper are clearly in the GraphML scope, **this paper does not pass the test: "Every submission must clearly state how the work is relevant to the Web and to the track in the first page"**.

- a full page of related work is in the supplementary material. It should probably be shortened to retain the most relevant information and integrate in the main text, to give the reader a better understanding of the context. This section is not announced in the current main text, and it looks like it was just put in supplementary to avoid overflowing beyond the page limit. Similarly, the dataset presentation (at least some figures) should be included in the main text.

- also, there is currently a lot of activity in designing efficient spatio-temporal models for learning, not specifically on the "traffic prediction" task. E.g., in the "link prediction" community (recommender systems, anomaly detection, etc.), spatio-temporal Transformer-based architectures are built (spatial does not necessarily means "geography" but more "topology"). There are very robust benchmarking frameworks (see Poursafaei, F., Huang, S., Pelrine, K., & Rabbany, R. (2022). Towards better evaluation for dynamic link prediction. Advances in Neural Information Processing Systems, 35, 32928-32941). It would have been great to compare the proposed model and evaluate it against some of those architectures.

Minor point:
- the Figure 1 does not provide information about the data used, nor the significance of points A, B, C, and D. Those points could have been specifically selected for exhibiting a behavior, aren't there more robust studies?

**Questions:**

- Could you detail more the complexity results/claims? Especially, it is implied that T<<N (but T's are still presented in the O() notation): perhaps the text should state more clearly what "linear" means (space, time), etc.

- Could you clarify (for the reviewer and in text) what you mean by "enables free interaction" and "each node interacts without complex temporal relationships […]"? It precisely seems to me that "free interaction across different timestamps and locations" will lead to "complex relationships"…

- Could you give some intuition regarding some parameters:  e.g., what does k=12 means in terms of traffic in a city? What nodes tend to be
"neighbors" with this DTW transformation?

**Reviewer Confidence:**

3: The reviewer is confident but not certain that the evaluation is correct

**Scope:**

1: The work is irrelevant to the Web